# Collagen Membranes Functionalized with 150 Cycles of Atomic Layer Deposited Titania Improve Osteopromotive Property in Critical-Size Defects Created on Rat Calvaria

**DOI:** 10.3390/jfb14030120

**Published:** 2023-02-23

**Authors:** Leonardo P. Faverani, Sarah Hashemi Astaneh, Monique Gonçalves da Costa, Leonardo A. Delanora, Tiburtino J. Lima-Neto, Stéfany Barbosa, Maretaningtias Dwi Ariani, Christos Takoudis, Cortino Sukotjo

**Affiliations:** 1Department of Diagnosis and Surgery, School of Dentistry, Sao Paulo State University (UNESP), Aracatuba 16015-050, Brazil; 2Department of Chemical Engineering, University of Illinois Chicago, Chicago, IL 60612, USA; 3Faculty of Dental Medicine, Universitas Airlangga, Surabaya 60132, Indonesia; 4Biomedical Engineering Department, University of Illinois Chicago, Chicago, IL 60607, USA; 5Department of Restorative Dentistry, University of Illinois Chicago, Chicago, IL 60607, USA

**Keywords:** atomic layer deposition, membrane functionalization, titanium dioxide, osseointegration

## Abstract

The membranes used in bone reconstructions have been the object of investigation in the field of tissue engineering, seeking to improve their mechanical strength and add other properties, mainly the osteopromotive. This study aimed to evaluate the functionalization of collagen membranes, with atomic layer deposition of TiO_2_ on the bone repair of critical defects in rat calvaria and subcutaneous biocompatibility. A total of 39 male rats were randomized into four groups: blood clot (BC), collagen membrane (COL), COL 150—150 cycles of titania, and COL 600—600 cycles of titania. The defects were created in each calvaria (5 mm in diameter) and covered according to each group; the animals were euthanized at 7, 14, and 28 days. The collected samples were assessed by histometric (newly bone formed, soft tissue area, membrane area, and residual linear defect) and histologic (inflammatory cells and blood cells count) analysis. All data were subjected to statistical analysis (*p* < 0.05). The COL150 group showed statistically significant differences compared to the other groups, mainly in the analysis of residual linear defects (1.5 ± 0.5 × 10^6^ pixels/µm^2^ for COL 150, and around 1 ± 0.5 × 10^6^ pixels/µm^2^ for the other groups) and newly formed bone (1500 ± 1200 pixels/µm for COL 150, and around 4000 pixels/µm for the others) (*p* < 0.05), demonstrating a better biological behavior in the chronology of defects repair. It is concluded that the collagen membrane functionalized by TiO_2_ over 150 cycles showed better bioactive potential in treating critical size defects in the rats’ calvaria.

## 1. Introduction

Alveolar bone remodeling is a constant physiological process which occurs after tooth loss resulting from caries, periodontitis, dentoalveolar trauma, pathological processes, malformations, or osteonecrosis caused by irradiation or use of antiresorptive/antiangiogenic drugs. The process is progressive, cumulative, and irreversible, being faster in the first six months, and continuous throughout the life of the patients. Such conditions may generate discomfort to the patient, due to functional, aesthetic, and psychological impairment, since social life tends to be affected [1,2,3,4,5,6]. There is an increasing number of individuals who seek oral rehabilitation through prostheses on osseointegrated implants, which need a sufficient amount of bone remaining to have functional stability and not harm adjacent tissues and structures [7,8].

In several cases, alveolar bone remodeling may cause different sizes of defects. Small defects, for instance, loosen the alveolar buccal wall after tooth extraction, starting after two months postoperatively, and showing more clinical relevance after six months [9]; alternatively, the even greater defects, those defined as critical, which loosen more than two bone walls may occur [9]. It is necessary to perform a bone reconstruction in those clinical situations, sometimes before the dental implant placement [10,11]. There are currently four categories of bone graft materials: autogenous (donor and recipient are the same individuals), homogenous (occurs between two individuals of the same species), heterogeneous (occurs between two individuals of different species), and alloplastic (graft material is of mineral or synthetic origin) [10,11]. The use and indication of the best material will depend on clinical applications, the volume of the deficiency, and evidence-based studies [11,12]. Autogenous bone grafts are considered the gold standard for bone rehabilitation since they have osteoconductive, osteoinductive, and osteogenic properties [13]. However, their use is associated with greater discomfort to the patient owning to the need for two surgical accesses, greater operative time, and greater risk of infections [13].

To reduce the morbidity and discomfort of patients undergoing bone reconstructions, studies have been carried out to develop new materials and biological treatments that can favor and accelerate the bone tissue repair process or even act as bone substitutes [14]. Collagen is the most abundant protein in the animal kingdom; moreover, it is one of the main elements present in different parts of the human body, mainly in bones and skin [15,16]. The use of collagen-based biomaterials has gained ground in different biomedical areas, especially in the field of tissue engineering, owing to their biocompatibility and biodegradability, emphasizing their osteopromotive property as membranes for guided bone regeneration [14].

The collagen membranes used for guided bone regeneration available on the market are derived from bovine and porcine type I and III collagen, originating from the Achilles tendon, dermal matrix, peritoneum, and pericardium. The use of collagen as a membrane is because it is transferable from animal to human, shows an active role in blood clot formation, and it promotes rapid wound stabilization [17]. The osteopromotive property using collagen membranes has been investigated and reported as a selective barrier that allows vascular cells’ and other osteoblastic cells’ migration; it is also biocompatible and biodegradable, and avoids epithelial cell migration [18]. Thus, the membranes should allow traffic of the vascular and osteogenic cells between soft tissue and the reconstructed area. That is possible, especially for non-resorbable membranes, where there are pores created with enough size to allow the necessary permeability [18].

Although they have excellent biological properties, collagen membranes have inadequate mechanical properties, low fibrillar density, and poor interaction with bone tissue due to their highly hydrated structures [19]. In an attempt to overcome these limitations, studies were carried out with the objective of incorporating bioactive compounds into the structure of collagen, such as the incorporation of bioactive glass, direct inclusion of hydroxyapatite, incorporation of ceramic compounds, and other nanomaterials [10,19,20,21,22,23]. Recent research has used titanium dioxide (TiO_2_) in this incorporation. Several techniques have been applied to prepare the surface of biomaterials with TiO_2_, including chemical vapor deposition, electronic beam evaporation, sputtering, pulsed laser deposition, and atomic layer deposition (ALD) [24,25,26,27,28].

Such controlled temperature deposition is already a well-established and advantageous coating method since it makes it possible to control the uniformity and thickness of the deposited films [24]. The use of atomic layer deposition specifically with TiO_2_ has been widely used due to its excellent properties such as biocompatibility, antimicrobial activity, hydrophilicity, excellent resistance to corrosion, aesthetics, low cost, and its non-toxic nature in contact with the human body [24]. An in vivo study demonstrated that the deposition of the TiO_2_ atomic layer obtained good results after application on the surface of titanium implants, giving nano-hardness and significant inhibition of the adhesion and growth of S. aureus and E. coli, bacteria frequently involved in infections of osseointegrated implants [24].

Recently, the functionalization of collagen fibers with 150 and 600 cycles of TiO_2_ deposition led to thicker collagen fiber diameters and more hydrophilicity compared to noncoated collagen fibers. The functionalized collagen fibers demonstrated a greater attraction for calcium and phosphate and improved biocompatibility, promoting higher growth and proliferation of human mesenchymal stem cells when compared to noncoated collagen fibers. These properties indicate that functionalization may have the ability to favor the formation of bone tissue [14]. It is important to perform in vivo experiments that support functionalized collagen membranes through the deposition of TiO_2_ and the biological behavior of this incorporation, since the literature on this topic is barely available. Therefore, based on previous literature [14], this in vivo study aimed to evaluate the osteopromotive behavior of functionalized collagen membranes using TiO_2_ applied by 150 ALD cycles and 600 ALD cycles in critical-size defects created on rat calvaria.

The null hypothesis of this study was that there would not be any difference among tested membrane groups (COL, COL150, and COL600), even in comparison with the control group (BC).

## 2. Materials and Methods

### 2.1. Sample Preparation and Characterization

In this work, atomic layer deposition (ALD) was used for all thin film depositions based on the previous publications [23,29,30,31].

Biomend^®^ absorbable collagen membrane (distributed by ZIMMER|dental, Warsaw, IN, USA), was used as collagen substrate. This membrane is composed of 100% type I collagen from Bovine tendon. Tetrakis (dimethylamido) titanium (IV) (TDMAT, San Luis, MO, USA) was used as the titanium precursor (Sigma Aldrich, San Luis, MO, USA). O_3_ was used as the oxidizer for this ALD and it was prepared using a UV-ozone generator placed immediately upstream of the deposition chamber to minimize ozone decomposition in the delivery line as described in our previous studies [29,30]. Ultra high purity nitrogen (N150 UH-T, Praxair Distribution Inc., Orlando, FL, USA) was used as a carrier gas and purging gas during all experiments.

In the current study, ALD of TiO_2_ thin films for the animal study was performed using the Kurt J. Lesker 150LE system near room temperature (~40 °C). Deposition pressure was ~1000 mtorr. To avoid precursor condensation on linings of the system, the designed recipe for room temperature TiO_2_ deposition had longer purging times in comparison to other thermal ALD of TiO_2_. Each cycle of the ALD reaction consisted of 1 s/50 s precursor pulse/purge and 1.8 s/45 s oxidizer pulse/purge. Silicon wafer (University wafer, Inc., South Boston, MA, USA) was used as a reference substrate to monitor the growth and thickness of the deposited TiO_2_ thin film in each batch. Three groups of samples were studied for animal testing: COL (pristine collagen), COL150 (collagen functionalized with 150 cycles of TiO_2_ ALD), and COL600 (collagen membrane functionalized with 600 cycles of TiO_2_ ALD).

Spectral ellipsometry (M44, J.A. Woollam Co., Inc., Lincoln, NE, USA) was performed on each cut-out of Si wafer for thickness measurements. The average thicknesses of TiO_2_ thin film on COL150 and COL600 were 9.1 ± 0.12 nm and 30.4 ± 2.72 nm, respectively. Surface chemical composition of samples was analyzed using Kratos AXIS-165, Kratos Analytical Ltd., Manchester, UK equipped with a monochromatic Al Kα X-ray source on large area mode (1150 μm × 70 μm). Samples were attached to the sample holder using double-sided carbon tape. During XPS, pressure of the chamber was maintained at less than 10-8 torr. XPS of the samples is presented in Figure 1 After functionalizing collagen membranes with different thicknesses of ALD TiO_2_, samples showed the characteristic peaks of Titanium, Ti 2s and 2p at 560 and 455 eV, respectively. These results were in line with the previously reported results in the literature [29,30] and confirmed the effective functionalization capability of ALD on biological samples.

### 2.2. Animal Study

#### 2.2.1. Ethics Statement

The project was submitted and approved by the Ethics committee for the use of ani-mals from the Sao Paulo State University (UNESP), School of Dentistry, Aracatuba—SP, Brazil (#411-2020), and followed the ARRIVE guidelines in animal studies [31].

Thirty-nine male, adult (6 months) Wistar rats (Rattus norvegicus), ranging from 250 to 300 g of weight, were used in this study. Thirty-six rats were used for critical size defect assay and the remaining three for biocompatibility assay.

The animals were originated and kept at the Vivarium of Aracatuba Dental School, four animals per cage, with controlled temperature (22 ± 2 °C) and light cycle (12 h of light and 12 h of dark), and provided solid food and water at libitum except within eight hours preoperatively.

#### 2.2.2. Sample Size

Data about new bone formed (“Primary outcome”) (mean difference = 18.9 and standard deviation = 8.1) and assessed from a previous study [32] were used to determine the sample size for this research. The sample should be at least five for alpha of 5%, and a power test of 80%. The “*n*” was considered the bone defect in rats calvaria.

### 2.3. Surgical Procedures

All surgical procedures were performed by the same surgeon (TJLN) to avoid surgical technique bias.

#### 2.3.1. Biocompatibility Assay (Subcutaneous Tissue Behavior around Membranes)

The same sedative and anesthetic protocols were applied for those animals. Trichotomy and antisepsis were made on the backs of the rats. Three incisions of 2 cm were performed on the back of the rats (two on the cranial region (one on the left region, and one on the right region), and one on the right caudal region). Thus, all animals received the three membranes assessed (COL—collagen; COL150—collagen membrane with 150 ALD cycles of TiO2; COL600—collagen membrane with 600 ALD cycles of TiO2), distributed aleatory. The membranes were stabilized in the subcutaneous tissue through a monofilament suture (Nylon 5.0, Mononylon, Ethicon, Johnson Prod., São José dos Campos, Brazil). This same material was used for skin suture. 

The same postoperative protocol was applied. These animals were euthanized at three postoperative days for biocompatibility analysis. When the samples were collected, the suture used for membrane stabilization was removed and the opposite area was considered for analysis.

#### 2.3.2. Calvaria Defect

Thirty-six animals were subjected to a bilateral calvaria critical size defect, and divided into four groups (three rats; six defects/per group/per time), as described below:Blood Clot—BC (Negative group)—*n* = 9 rats: the bone defects were filled only by the blood clot; 3 rats were euthanized in each period of analysis (7, 14, and 28 days after surgery);Collagen Membrane—COL (control group)—*n* = 9 rats: the bone defects were covered by collagen membrane; 3 rats were euthanized in each period of analysis (7, 14, and 28 days after surgery);Collagen Membrane with 150 atomic layer deposition cycles of TiO_2_—COL150 (test group 1)—*n* = 9 rats: the bone defects were covered by membrane with 150 atomic layer deposition cycles of TiO_2_; 3 rats were euthanized in each period of analysis (7, 14, and 28 days after surgery);Collagen Membrane with 600 atomic layer deposition cycles of TiO_2_—COL600 (test group 2):—*n* = 9 rats: the bone defects were covered by membrane with 600 atomic layer deposition cycles of TiO_2_; 3 rats were euthanized in each period of analysis (7, 14, and 28 days after surgery).

The three remaining animals received the three membranes analyzed (COL, COL150, and COL600) subcutaneously for biocompatibility assay.

The randomization of the animals for assay (Critical size defect—36 animals—or Biocompatibility—3 animals) and groups (BC, COL, COL150, and COL600 for critical size defect assay) was performed through an envelope system in the preoperative time.

##### Critical Size Defect Assay

The animals were sedated using intramuscular ketamine (50 mg/kg; Dopalen, Nova Alvorada, Brazil) and xylazine (5 mg/kg; Xilazin, Votuparim, Brazil), complemented with local anesthesia with Mepiadre 100^®^ (DFL, Taquara, Brazil). Trichotomy was performed on the skull of rats, then antisepsis with Degermant Polyvinyl Pyrrolidone Iodine (PVPI) and topical PVPI, followed by a V-incision with flap apex located in the frontal region. After that, a 5 mm diameter critical size bone defect was performed on each side of the parietal bone, through the bone tissue, up to the safe limit to maintain the integrity of the dura mater.

Subsequently, both bone defects were recovered following the experimental groups (BC, COL, COL150, and COL600). In sequence, the flap was carefully repositioned and sutured with interrupted monofilament suture (Nylon 5.0, Mononylon, Ethicon, Johnson Prod., São José dos Campos, Brazil). After the surgical procedure, all animals received a single intramuscular dose of 0.2 mL of Pentabiotic^®^ (Benzathine Benzylpenicillin 600.000 IU, Procaine Benzylpenicillin 300.000 IU, Potassium Benzylpenicillin 300.000 IU, Dihydrostreptomycin base (sulfate) 250 mg, and Streptomycin base (sulfate) 250 mg) (Pentabiótico Veterinário Pequeno Porte, Fort Dodge Saúde Animal Ltd.a., Campinas, SP. Every two days, the cages were cleaned, and the animals cared for.

These animals were euthanized at 7, 14, and 28 postoperative days. At 7 and 14 days, they were analyzed for histology/histometric assessment. At 28 days, they were analyzed for microtomography and histology/histometric assessment. 

### 2.4. Analysis

After euthanasia, the samples were collected (calvaria for critical-size defects analysis, and subcutaneous tissue for biocompatibility analysis), maintained in formaline for 48 h, and passed for laboratory process to include in paraffin. In this process, the samples are prepared so that the regions of interest face the external face of the paraffin blocks, so that they are microtomized perpendicularly. At 28 days postoperatively, the calvaria samples were scanned by computed microtomographic analysis. The biocompatibility and inflammatory assays (subcutaneous and calvaria defect) were performed by the same examiner (MGC). The histometric analysis was performed by another examiner (SB). All of them were blinded since all samples had their identification covered. 

#### 2.4.1. Biocompatibility Analysis (Subcutaneous Tissue Behavior around Membranes)

Each animal provides three pieces (one of each group). The histological slices were microtomized in 5µm and stained with Hematoxylin and Eosin. A camera (LeicaR DC 300F microsystems Ltd., Heerbrugg, Switzerland) coupled with an optical microscope (LeicaR DMLB, Heerbrugg, Switzerland) was used to obtain photomicrographs, which were analyzed in the ImageJ software (National Institutes of Health, Bethesda, MD, USA).

Regarding the inflammatory response, firstly, all samples were assessed to identify what type of cells were present. No signals of infection or foreign body reactions were noticed, represented by neutrophilic activity or multinucleated cells. The lymphocytes were fixed, and they were chosen for this analysis.

All images were imported to the ImageJ software using a grid tool with 130 points to perform the cell counting. Therefore, all points that crossed those cells were counted as the number of cells to compare among groups.

#### 2.4.2. Critical Size Defect Assay

##### Inflammatory Cells and Blood Cells Count

The histological bladess were stained with hematoxylin and eosin (Merk & Co., Inc., Kenilworth, New Jersey, NJ, USA). The inflammatory response was determined through qualitative analysis in which all histological slices did not show any different cells related to the tissue response, with more emphasis in lymphocytes.

One histological blade from each animal for the experimental time was chosen and two sections photographed under an image processing system, which consisted of a light microscope (DM 4000B, Leica), a color image processor (Leica Qwin V3, Leica software), a color camera (DFC 500, Leica), and a computer (Intel Core I5, intel Corp, Santa Clara, CA, USA; Windows 10, Microsoft Corp, Redmond, DC, USA) connected with ImageJ digitized image analyzer software (National Institutes of Health, Bethesda, MD, USA).

All samples had their identification hidden, so the examiner was blind. Three regions were assessed: the first image was taken in the center of the defect, followed by the one on the right and one on the left, totaling 72 images per experimental time/per group, at a ×100 magnification. All images were imported to the ImageJ software using a grid tool with 130 points that allowed enough points to perform the cell counting. Therefore, all points that crossed those cells were counted as numbers of cells to compare among groups.

##### Computed Microtomographic (Micro-CT)

Samples obtained at 28 days were used for structural characterization through mi-cro-CT. A total of five calvaria of each group were removed, maintained in 70% alcohol, and scanned with a SkyScan microtomography device (SkyScan 1176 Bruker MicroCT, Aatselaar, Belgium, 2003), under the following parameters: 17.4 mm^3^ voxel size, 8 µm sections, 90 Kv, 111 μA, with copper and aluminum filters, and a 0.05 mm rotation pitch. The images were reconstituted with NRecon software (SkyScan, 2011; Version 1.6.6.0; Bruker, Belgium) to determine the area of interest. Image reconstruction and position were performed in the Data Viewer software (SkyScan, Version 1.4.4 64-bit; Bruker, Belgium). After that, the CT-An software (SkyScan, 2012 Bruker MicroCT, Version 1.12.4.0, Bruker, Belgium) was used to define the interest areas (bone defect), which were analyzed separately. This area was defined as a 5 mm circle from the center of each bone defect (each defect is considered a sample). The software applies analysis in grayscale, known as threshold. For this study, the threshold was between 41 and 180. Then, the analysis was performed to evaluate the bone qualitative (Tb.Th = bone trabecular thickness, Tb.SP = separation of bone trabeculae, and Tb.N = number of trabeculae) and quantitative (BV.TV, = percentage of bone volume) parameters.

#### 2.4.3. Histometric Analysis

##### New Bone Formed (NBF)

As processed for inflammatory cells and blood cells count, for newly formed bone, all samples were subject to photomicrography at a ×6.3 magnification (DM 4000B, Leica, Munich, Germany). The central area of the bone tissue of each bone defect was evaluated using two sections per animal (primary outcome). On average, 18 pictures were taken for each defect, and then those images were transferred to the Adobe Photoshop CC 2019 to merge in a panoramic view. All the images were transferred to the ImageJ program, and using the tool “freehands”, the newly formed bone was measured considering the central region of the calvaria defect.

##### Soft Tissue Area

For analysis of the soft tissue area, the images were captured and analyzed in the ImageJ program version 1.52a (Image Analyzer Program, Toronto, ON, Canada), in which the histometric calculation was performed. For this, the program was calibrated using the “set scale” tool, using pixels as the standard measure. Then, the images were opened in the program and using the “freehands” tool, the soft tissue area was calculated. In situations where areas of interest for measurement were separated, the measurement was obtained by performing the sum of these areas.

##### Membrane Area

The captured images were recorded in a TIF file and analyzed in the ImageJ program, already mentioned, where the histometric calculation was performed. For this, the program was calibrated using the “set scale” tool, using pixels as the standard measure. Subsequently, the images were opened in the program and using the “freehands” tool, the area in pixels was calculated. In situations where areas of interest for measurement were separated, the measurement was obtained by performing the sum of these areas.

##### Residual Linear Defect

The captured images were also recorded in a TIF file, followed by the assembly of panoramic images in Photoshop CC 2019 version, and later analysis was performed by the ImageJ program to determine the histometric calculation. For this, the program was calibrated using the “set scale” tool, using pixels as a standard measure. Then, the images were opened in the program, and using the “straight” tool, measurement was performed in a linear manner.

##### Statistical Analysis

All tests were performed using the SigmaPlot 12.0 software (Exakt Graph and Data analysis Inc., San Jose, CA, USA). Initially, the data obtained from the histometric analysis underwent the normality test (Shapiro–Wilk). After that, the analysis of variance Two-Way ANOVA and post-Tukey tests were applied to compare the experimental groups (four levels: BC, COL, COL150, COL600) vs. periods (three levels: 7 days, 14 days, 28 days) for blood cells count, NBF, soft tissue area, and residual linear defect. For the Membrane Area, the post-test applied was Student–Newman–Keuls. For inflammatory cells count in the calvaria defect, the Kruskal–Wallis test was performed. A One-Way ANOVA test was applied to compare the experimental groups related to the micro-CT parameters (BV/TV, Tb.Th, Tb.Sp, and Tb. N). All tests considered a significance level of *p* < 0.05.

## 3. Results

### 3.1. Biocompatibility Analysis (Subcutaneous Tissue Behavior Around Membranes)

The membrane behavior in the subcutaneous tissue 3 days postoperatively showed an inflammatory response with no macrophages indicating a foreign body reaction. A few lymphocytes and blood vessels were noticed (Figure 2A), with soft organized tissue in all analyzed samples. A similar inflammatory cell counting was found for all groups (*p* > 0.05; Figure 2B).

### 3.2. Inflammatory Cells and Blood Cells Count (Calvaria Defect)

The inflammation on the calvaria during the bone healing showed higher lymphocytic cells at the beginning of analysis up to day 14 (Figure 3A,B), and significantly decreased on day 28 (*p* < 0.05). The tested membranes (COL150 and COL600) were the groups with higher numbers of inflammatory cells in comparison with the BC and COL groups at 7 and 14 days (*p* < 0.05). At 28 days, all groups showed similar data (*p* > 0.05). 

The connective tissue became more organized on day 14 in all groups, and the tested groups (COL150 and COL600) had more blood vessels in all samples (Figure 3A). Those data were confirmed by the blood vessels count, in which COL150 showed the highest count, and the COL group had the lowest count at 7 and 14 days of analysis (*p* < 0.05) compared with all other groups (Figure 3C). 

### 3.3. Panoramic Histological View and Tridimensional Microtomography Assessment (Calvaria)

The histological samples presented different levels of newly formed bone, filling the bone defect according to the analyzed groups (Figure 4A). BC only showed a small amount of new bone at the central area of the defect, and a thin layer of connective tissue filled the defect until day 28. All membranes (COL, COL150, and COL600) maintained their structure covering the defect until day 28, with no signs of collapsing into the bone defect. COL150 showed a higher extension of defect filling with newly formed bone in comparison with other groups, especially at 28 days postoperation (Figure 4A).

Regarding the microCT assessment, for the BV/TV parameter, BC showed the lowest values compared with the other groups (*p* < 0.05; Figure 4B). The quality of newly formed bone did not show any difference among group tissues (Tb.Th, Tb.Sp, and Tb. N; *p* > 0.05). 

### 3.4. New Bone Formed (NBF) (Calvaria)

At 7 and 14 days, all experimental groups (BC, COL, COL150, and COL600) showed similar data for NBF (*p* > 0.05; Figure 4C), despite all membranes numerically showing a numerical increase on day 14. At 28 days, COL150 presented the highest value for NBF, followed by COL600, COL, and BC (*p* < 0.05; Figure 4C).

### 3.5. Soft Tissue Area (Calvaria)

At 7 days, the BC group showed the highest amount of soft tissue area (*p* < 0.001; Figure 4D), and the other groups (COL, COL150, and COL600) showed similar data in the comparison among them (*p* > 0.05). The BC and COL groups decreased the amount of soft tissue from 7 to 14 days postoperatively (*p* < 0.05). COL150 and COL600 had a similar amount of soft tissue in all analyzed periods (*p* > 0.05).

### 3.6. Membrane Area (Calvaria)

The membrane area demonstrated that COL600 had a lower value in all analyzed periods (*p* < 0.05; Figure 4E). Only the COL group decreased from 7 to 14 days and from 14 to 28 days postoperatively (*p* < 0.05). On day 28, COL and COL150 presented similar values (*p* = 0.35), and both values were higher than COL600 values (*p* < 0.05). 

### 3.7. Residual Linear Defect (Calvaria)

The analysis focused on the residual linear defect on day 28, demonstrated lower values for COL150 (*p* < 0.01), and all other groups had similar values (BC = COL = COL600; *p* > 0.05) (Figure 4F).

## 4. Discussion

The use of critical defects in rodent calvaria is a well-established experimental animal model widely used in the scientific community to evaluate bone remodeling process, osteopromotive and osteoconductive properties of materials with potential for guided bone regeneration [13,33,34,35,36,37] and it was the model chosen for this study. The results of this study clearly showed that the COL150 membrane yielded better biological behavior during the chronology of bone healing of defects created in rat calvaria, with an emphasis on the primary outcome, the area of newly formed bone between the remaining bone stumps. The COL600 membrane did not promote additional effects on bone healing compared to the other groups tested, suggesting the filling of the bone defect with Ti ions until the last period of analysis, providing the formation of new bone between the particles, occupying the spaces created by the membrane, which was slowly degrading.

In this study, the deposition of metal oxide coatings, such as TiO_2_, can be considered a beneficial option adopted in improving the properties of biomaterials, as it has been investigated [38]. The literature suggests that incorporating metal oxides from the deposition of the atomic layer is becoming a promising process for adapting properties such as hydrophilicity and hydrophobicity that are characteristic of biomedical surfaces. In addition, the surface treatment inhibits the growth of microorganisms, stimulates the process of cell proliferation and differentiation, and improves the biomechanical properties of these materials, such as the osteopromotive membranes [14,24,38].

For the development of membranes or any other biomaterials for clinical application, the analysis of the behavior of these biomaterials in a physiological organism is essential, and the inflammatory profile is an important parameter to identify if there is biocompatibility in the soft and bone tissues [39,40]. Previous in vitro studies and cell cultures have shown biocompatibility and significant osteogenic growth by ALD using titanium oxide coating and other film deposition [27,28,29,38,39,41]. As the first tissue response on the bone healing is inflammation, this study assessed the membranes’ subcutaneous behavior at days postoperatively in the rats’ dorsum, and on the calvaria bone defects at 7, 14, and 28 days postoperatively.

It was clear that despite a greater number of lymphocytes for both tested collagen membranes (COL150 and COL600) at the beginning of bone healing (day 7), TiO_2_ deposition did not stimulate tissue reactions with the invasion of macrophage cells in any of the experimental groups and times, including in the most accurate analysis on the ×100 objective (Figure 2A). Furthermore, in the subcutaneous assessment, all membranes showed the same behavior, confirming their biocompatibility in an in vivo model. 

For the biocompatibility analysis, the group used as a control was the COL, as also used in previous publications [18,19,42]. The tested membranes for subcutaneous assessment had the same standardized surgical approach. Therefore, this study did not consider a group with only a blood clot trauma from the surgery for not providing any substantial data.

The three-dimensional microtomographic parameters showed that all membranes tested and compared in this study showed similar behavior for the quantity and quality of bone tissue, with all comparisons being higher than the BC group. These data corroborate findings in the literature that report the use of collagen membranes as a good option to accelerate and assist the bone regeneration process, which is associated with its chemotaxis capacity for fibroblasts, acting as a physical barrier for endothelial cell migration, biocompatibility, and biodegradation. In addition, this type of biomaterial has a permeable selective system allowing the traffic of vascular and osteogenic cells between soft tissue and the reconstructed region, thus resulting in more satisfactory repair processes than in groups where the defect was repaired only by the physiological mechanism [18,19].

Regarding the two-dimensional histometric chronology through HE stained histological sections, with evidence of bone tissue neoformation values, in the first two experimental periods (7 and 14 days), all groups studied presented similar results; however, at 28 days, the COL150 group increased the values of newly formed bone compared to the other groups. These results demonstrated that the membranes modified by the deposition of TiO_2_ showed excellent osteopromotive properties when compared to the groups COL and BC for example, proving that when a biomaterial has its surface modified in suitable conditions, its properties are improved. It is even possible to add osteoconductive characteristics of the incorporated material, as in the case of TiO_2_. The difference between the highest regenerative values of the COL150 group from those with the COL600 group, showed that although the incorporation of TiO_2_ is beneficial, for COL600, the increase in the number of cycles was excessive, which promoted the filling of the critical defect by the incorporated material, with fewer gaps to allow new bone formation. Bishal et al., 2017 [23] found that 600 cycles of TiO_2_ deposited on the collagen membranes increased the fiber membranes’ diameter by more than 200 nm compared with 150 cycles (around 60 nm); for this experimental design using a critical-size defect, COL600 had more of a barrier behavior.

Hashemi Astaneh et al., 2020 [41] reported in the study of osteoblastic cell culture that the increase of 600 cycles of TiO_2_ in the collagen membrane significantly increased the attractiveness for osteogenic cells and with greater maturation in the analyzed periods. However, in that study design, there is no physiological component being evaluated, such as neoangiogenesis, which is fundamental for the behavior of resorbable membranes in bone reconstructions, which given the presence of pores on its surface, allows the selectivity of cells of interest to the region that is in regeneration. Therefore, the histometric data clarify that the COL600 group did not behave in the same way as in previously published studies. The data from this study were fundamental to confirm the necessity for animal studies before clinical investigations in assessing the development and properties of the biomaterial and starting to think about its commercialization.

The incorporation of bioactive substances to the osteopromotive membranes has aims besides the increase of mechanical resistance of the biomaterial. In terms of three-dimensional reconstructions, the membranes need to be stabilized and need to ensure that the scaffold is maintained for a considerable period until the minimum amount of immature bone has been deposited to maintain the desirable reconstructed bone volume. These investigations also aim to include the osteoconductive property into the membranes, since coated with TiO_2_ they are more bioactive. They promoted a higher rate of osteoblast growth, propagation, and cell proliferation, as well as the calcium phosphate nucleation or formation of apatite. These results were attributed to the effectiveness of the coating on the collagen surface in inducing a higher gene expression level, thus increasing the capacity for bone formation compared to uncoated control groups [14,24,41,43,44,45].

The analysis of the resorption dynamics of the membrane area is an important parameter to verify the compatibility pattern of the resorbable membranes and the potential for contribution to the bone regeneration process. COL and COL150 membranes showed the same behavior at all experimental times, with higher residual linear defect values compared to the COL600 group. This confirmed the findings previously observed in histometric analysis that the 600 cycles of TiO_2_ ended up not presenting results as satisfactory as the COL150 group for osteopromotive effects, due to the amount of incorporated titanium dioxide being excessive. This resulted in an obstruction (filling) of the critical defect by the incorporated material and, consequently, there was no degradation of the collagen membrane with replacement by newly formed bone. These results corroborate the study by Silva et al., who reported that the deposition of metal oxide coatings further improves the performance of the biomaterial in question, provided that it is carried out properly and in ideal quantities [38] since the excessive incorporation of metal oxides by the ALD technique ends up delaying the tissue repair process.

Regarding the limitations of this study, it is difficult to identify the inflammatory phase and its relationship with the initial response related to bone formation. Although the quantification was performed by a calibrated researcher and focused on the number of leukocytes, it is difficult to determine whether the noticed inflammation refers to a controlled pro-inflammatory response or shows an exacerbated inflammatory activity. Despite that, this limitation is overcome when the inflammation data leads to a good amount of newly formed bone in the long-term analysis, as noticed in this study. This shows that the initial inflammatory response was controlled and allowed the next phase of bone healing. 

Another limitation of the study was the disposition and magnitude of the three-dimensional bone defect, which did not make it possible to analyze micro-angiogenesis during bone regeneration due to the pores in the collagen membranes. In future studies, micro-angiographic analysis may be carried out to investigate and confirm the current data collected. Additionally, the increase in mechanical strength by 150 cycles of TiO_2_ ALD (COL150) promotes the proposal to test the stabilization of membranes in larger three-dimensional defects for vertical and horizontal bone augmentation. As the study showed the biocompatibility of the COL150 membrane, future studies should investigate the regenerative principles of this membrane clinically to confirm and expand the findings of this study and its applicability in the clinical routine.

## 5. Conclusions

Within the limits of this in vivo study, it can be concluded that functionalized collagen membranes with 150 cycles of ALD TiO_2_ showed the greatest osteopromotive property in critical-size defects on the calvaria of the rats.

## Figures and Tables

**Figure 1 jfb-14-00120-f001:**
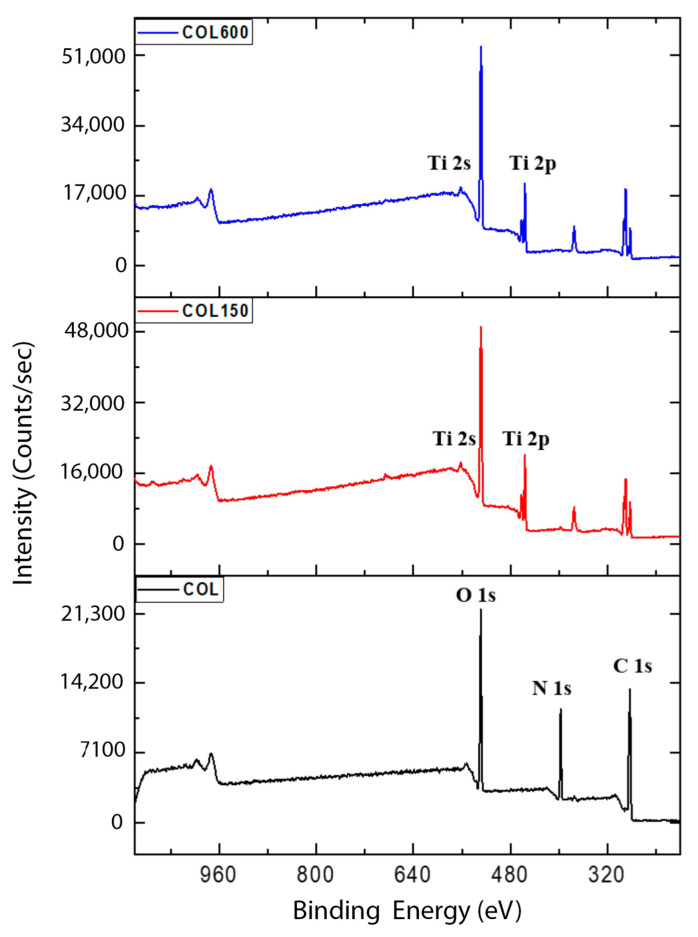
XPS spectra on collagen membrane (COL), 150 ALD cycles of TiO_2_ (COL150), and 600 ALD cycles of TiO_2_ (COL600). All depositions were done at near room temperature, ~40 °C, 1000 mtorr, and 1 s/50 s precursor pulse/purge, 1.8 s/45 s oxidizer pulse/purge in a Kurt J. Lesker 150LE ALD system.

**Figure 2 jfb-14-00120-f002:**
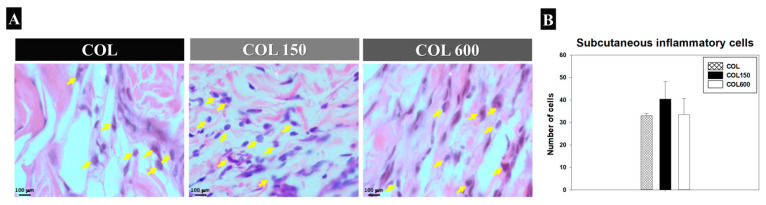
(**A**) Photomicrograph of the histological sections of the experimental groups (COL, COL150, and COL600) showing the subcutaneous inflammatory cells three days postoperatively. The inflammatory cells are indicated by yellow arrows. (HE staining; Objective ×40) (**B**) Representative graph of the means and standard deviations from the subcutaneous inflammatory cells count of the experimental groups (COL, COL150, and COL600) 3 days postoperatively.

**Figure 3 jfb-14-00120-f003:**
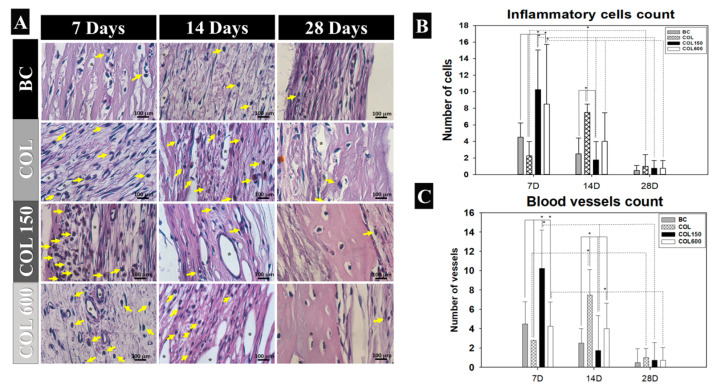
(**A**) Photomicrograph of the histological sections of the experimental groups (BC, COL, COL150, and COL600) showing the inflammatory cells and blood vessels at 7, 14, and 28 days after surgery. Inflammatory cells are indicated by yellow arrows and blood vessels are indicated by black asterisks. (HE staining; ×40 objective). (**B**) Representative data of mean and standard deviation from the inflammatory cells count in the calvaria repair at the times of 7, 14, and 28 days after surgery of the experimental groups (BC, COL, COL150, and COL600) (* *p* < 0.05). (**C**) Representative data of mean and standard deviation from the blood vessels count of the experimental groups (BC, COL, COL150, and COL600) according to time (7, 14, and 28 days postoperatively) (* *p* < 0.05).

**Figure 4 jfb-14-00120-f004:**
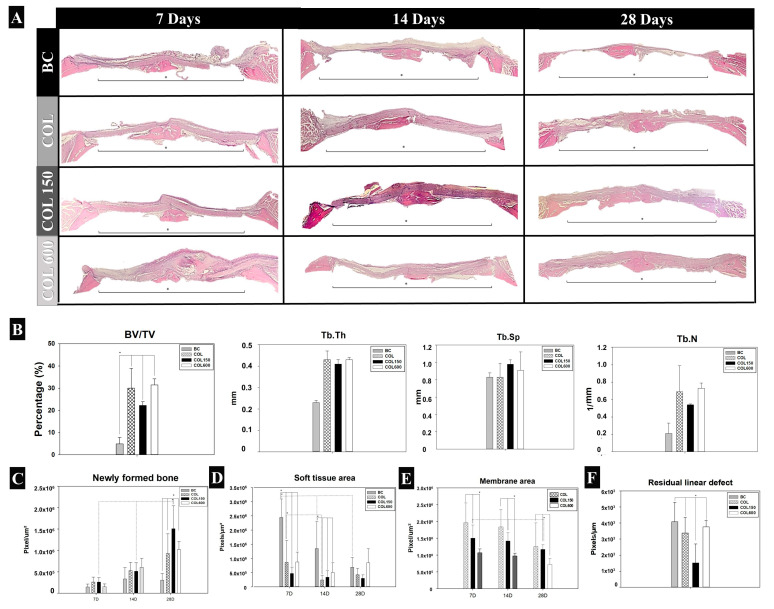
In vivo data. (**A**) Representative panoramic images of the histological assessment for all experimental groups (BC, COL, COL150, and COL600) 7, 14 and 28 days postoperatively. The asterisk represents the center of the defect, and the black line delimits the area of the bone defect created. (HE staining; x6.3 objective). (**B**) Representative means and standard deviations of the micro CT parameters showing the values related to amount of bone tissue (BV/TV) and quality of bone tissue (Tb.Th, Tb.Sp, and Tb. N) at 28 days after creation of critical-size defects on rats calvaria and coverage by membranes (COL, COL150, and COL600), compared with BC. * Only in the comparison among COL, COL150, and COL600 versus BC was there a value with a significant difference (*p* < 0.05) for BV/TV. Graphs of the histometric analysis of newly formed bone (**C**), soft tissue area (**D**), residual membrane area (**E**), and residual linear defect (**F**) of the experimental groups (BC, COL, COL150, and COL600) based on time (7, 14, and 28 days postoperatively). ((*) *p* < 0.05).

## Data Availability

All research data are kept in a restricted drive and are unavailable owing to patent restrictions.

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
