# Peer review of "Collagen Membranes Functionalized with 150 Cycles of Atomic Layer Deposited Titania Improve Osteopromotive Property in Critical-Size Defects Created on Rat Calvaria"

_jfb, 2023, doi:10.3390/jfb14030120_

Round 1
Reviewer 1 Report
Dear authors,
thank you for the opportunity to review the interesting manuscript. There are major points that need adaptation, especially regarding the description of the experimental procedures:
Abstract:
L.17: The term "functionalization" is unclear, please add a definition or praphrasation.
In the abstract it is unclear which analysis methodologies were used in the study. Please be more concise about the specific tests.
Introduction:
L. 37: Do you mean first six months after tooth loss? Please specify and maybe add a reference.
L. 45: Please add a reference.
L. 54: Please add at least one reference.
L. 75: Please provide a definition for "bioactive".
L. 78 etc.: Please use the correct chemical formatting for molecules as TiO2 or O3 with a subscripted number wherever applicable in the whole manuscript.
Please name at least one null-hypothesis at the end of your introduction.
Mat/Met:
L. 103:
At the beginning of the Mat Met section, you should again define all abbreviations when first used.
Additionally, in the article there is no discrimination regarding collagen types. You should be consistent regarding the collagen type used in your study and described in your introduction. Please provide some more informations about the membrane composition.
L. 105 etc. : Please provide the company and the country of origin for all materials directly after first mention as it is outlined in the submission guidelines.
L. 162: Were the membrane-placing locations chosen randomly?
Did you include a control in the biocompatibility assay as well? (eg incision without membrane or only with blood clot as in the calvaria defect assay) If not, please argue in the discussion why.
L. 196: Formulation "5 mm critical size bone defect" is unclear. Please provide all size informations (width, length, volume)
L. 202: Please provide the exact composition of "Pentabiotic".
L.221-222: What is the definition of "inflammatory cells"? The sentence "with emphasis in lymphocytes" also suggests that it might be a rather qualitative criterion. "Please alsop provide some information about the "semi-quantitative analysis".
L. 235: How many surgeons and how many examiners performed the experiments.
L. 240: Please provide the µCT voxel size in this paragraph.
L. 258: Please rephrase this sentence grammatically.
In the Mat/Met-part, there are not enough informations regarding the blood vessel and cell counts. Were they counted per a specific area? Please add.
Results:
In general it is not clear why no µCT images are shown.
Fig. 3: This graph and its error bars with quite differing heigth suggest that data may not be normally distributed and should thus be treated non-parametrically in statistical analysis.
Discussion:
L. 382:
"showing the filling of the bone de-382 fect Ti ions until the last period of analysis, "
This sentence is unclear and it suggests that you investigated Ti ion distribution in your study, which was not the case.
L. 403: How did you make sure that you see lymphocytes in your study? You should critically discuss the limitaitons of your study regarding cell identification.
L. 419: You should critically discuss, how you made sure to make that your sections are perpendicular to the defects, which is a necessary prerequisite for metrical investigation.
Author Response
All revised points are described in the attachment below.

Reviewer 2 Report
In the study, the authors aimed to evaluate the functionalization of collagen membranes, with atomic layer deposition of TiO2 on the bone repair of critical defects in rat calvaria and subcutaneous biocompatibility. They tested the materials in a calvaria defect model in rates. They showed that the collagen memebrance with 150 cycles of depoistions of TiO2 showed a better biological behavior in the chronology of defects repair, suggesting the promises of t the collagen membrane functionalized by TiO2 with 150 in treating critical size defects in the rats calvaria. This study seems to be a follow up study with their previous study (Ref.13) demonstrating the material’s effectiveness in an animal model.
Comments:
1. As the materials used in this study were reported previously and the results from this study might be important for future translation, but the study itself lacks significant novelty.
2. In the introduction, the authors focus on the problem of Alveolar bone resorption by using two paragraphs, but they tested their materials using a calvaria defect model which is not appropriate to solve the problem of Alveolar bone resorption.
3. Althougth some parameters had been shown previously, it is important to include some data of material characterization, such as surface morphology, roughness and wettability measurements, which then might be used for discussion.
4. In Figure 2, the inflammation cell quantification does not match the image for Col150 group, which seems to be at least double than other two groups.
5. More discussions are need to address the reasons of choosing 150 and 600 cycles, and other groups (i.e. less than 150, or between 150 to 600) might be more effective? Which properties elicited by 150 cycles of deposition resulted the positive outcome?
6. Figures’ quality can be improved for better visualization.
Author Response
All revised points are attached below.

Round 2
Reviewer 1 Report
Dear authors,
we want to thank the authors for taking my comments on board. There are only few points left from my side:
General:
he revised version does not contain line numbers anymore. PÜlease add for an easier review process.
Abstract:
Except for p-values, the Abstract still does not contain numerical data. Please add.
Introduction:
"teeth loss" should be replaced by the more common "tooth loss".
"aggressive" is in my opinion not an adequate term for a physiological process as bone remodeling.
Mat/Met:
You added the voxel size. However, it is still unclear why your µCT-results are depicted as "Pixels/µm^3", as pixel is a 2d-unit.
Author Response
All revised points are attached below.

Reviewer 2 Report
It has been significantly improved after revision and can be accepted now.
Author Response
Thank you for your input. All text was revised to correct some typing errors and improve the text.